# Local Recurrences in Rectal Cancer: MRI vs. CT

**DOI:** 10.3390/diagnostics13122104

**Published:** 2023-06-17

**Authors:** Giulia Grazzini, Ginevra Danti, Giuditta Chiti, Caterina Giannessi, Silvia Pradella, Vittorio Miele

**Affiliations:** Department of Emergency Radiology, University Hospital Careggi, Largo Brambilla 3, 50134 Florence, Italy; ginevra.danti@gmail.com (G.D.); giudittachiti@gmail.com (G.C.); giannacate@tiscali.it (C.G.); pradella3@yahoo.it (S.P.); vmiele@sirm.org (V.M.)

**Keywords:** rectal cancer, recurrence, computed tomography, magnetic resonance imaging, functional imaging

## Abstract

Rectal cancers are often considered a distinct disease from colon cancers as their survival and management are different. Particularly, the risk for local recurrence (LR) is greater than in colon cancer. There are many factors predisposing to LR such as postoperative histopathological features or the mesorectal plane of surgical resection. In addition, the pattern of LR in rectal cancer has a prognostic significance and an important role in the choice of operative approach and. Therefore, an optimal follow up based on imaging is critical in rectal cancer. The aim of this review is to analyse the risk and the pattern of local recurrences in rectal cancer and to provide an overview of the role of imaging in early detection of LRs. We performed a literature review of studies published on Web of Science and MEDLINE up to January 2023. We also reviewed the current guidelines of National Comprehensive Cancer Network (NCCN) and the European Society for Medical Oncology (ESMO). Although the timing and the modality of follow-up is not yet established, the guidelines usually recommend a time frame of 5 years post surgical resection of the rectum. Computed Tomography (CT) scans and/or Magnetic Resonance Imaging (MRI) are the main imaging techniques recommended in the follow-up of these patients. PET-CT is not recommended by guidelines during post-operative surveillance and it is generally used for problem solving.

## 1. Introduction

Colon-rectal cancer is the third cause of cancer-related death in the United States, while in Europe it is the third most frequent cancer in men and the second in women [1,2]. Rectal cancer (RC) accounts for approximately 30% of colon cancer cases and it’s often considered as a distinct disease. Rectal cancers differ from colon cancers in survival and management. There are issues unique to rectal cancer, such as sphincter preservation and sexual dysfunction secondary to autonomic neuropathy. Moreover, surgical resection of the rectum is technically challenging, making the risk of local recurrences (LRs) greater than in colon cancer [3,4]. According to the National Comprehensive Cancer Network (NCCN) Guidelines, the total mesorectal excision (TME) is the gold standard for curative rectal surgery preventing the dissemination of tumour cells and reducing the risk of LR [4]. In addition, neoadjuvant chemoradiotherapy (CRT) followed by TME is recommended for locally advanced rectal cancers to reduce the risk of LR [4]. An alternative approach is the “watch and wait” strategy that can be indicated to patients with clinical complete response after neoadjuvant CRT. This conservative strategy showed a local recurrence rate of 25% and distant metastasis rate of 8% after 3 years as described by the International Watch and Wait Database [5].

LRs still represent a significant clinical problem causing debilitating symptoms such as bleeding, pelvic pain, and fistula. In addition, LRs are associated with high morbidity after salvage therapy. Therefore, surveillance programs based on imaging are critical for an early diagnosis of LRs [6]. Although the timing of follow-up is not yet established, about 95% of relapses occur within 5 years after surgery. For this reason, most guidelines recommend a follow-up of 5 years based on clinical examination and imaging [7]. The European Society for Medical Oncology (ESMO) guidelines suggest follow-up/surveillance up to 5 years after surgery, except for high-risk patients with positive circumferential resection margin (CRM) that deserve longer surveillance for LR. ESMO guidelines recommended clinical assessment, routine monitoring of carcinoembryonic antigen (CEA), colonoscopy, and Computed Tomography (CT) scans and/or Magnetic Resonance Imaging (MRI). In the follow-up, the use of PET-CT is only recommended for problem solving [8]. In accordance with the NCCN guidelines, surveillance involves history, physical examination, CEA test, endoscopyand chest, abdominal and pelvic CT scan in patients with stage II–IV [4].

The aim of this review is to analyse the risk factors and the pattern of local recurrences in rectal cancer and to provide an overview of the role of imaging in early detection of LR In particular, the purpose of this review is to evaluate which imaging method can best identify early local recurrence in rectal cancer by analyzing the recent literature.

## 2. Recurrences: Rate, Risk Factors and Pattern

Recurrence is defined as locoregional when there is clinical, radiological or anatomopathological evidence of tumour of the same histological type in the pelvis, at the site of anastomosis or in the mesentery up to the root of the inferior mesenteric artery, [3]. According to various studies, recurrent rectal cancer occurs in approximately 25–40% of patients undergoing surgery performed with curative intent, although the advent of neoadjuvant chemotherapy and improved surgical techniques have lowered the rate of recurrences in past years [9]. In their study on 326 patients with rectal cancer, Tan et al. found that, after 10 years of follow-up, about 8% of patients showed locoregional recurrences and about 22% had distant metastasis [3]. Rasanen et al. confirmed these data in their study on 481 patients with rectal cancer describing LRs in 8.3% of patients operated and distant metastases in 23.3% after 5 years of follow-up [9]. Recurrences may occur in the first 5 years of follow-up with a few cases (a little more than 1%) after the first 5 years [3,9,10,11].

There are many factors predisposing to LR of rectal cancer and among these the CRM and the mesorectal plane of surgical resection are two important prognostic factors. A positive (infiltrated) CRM is an independent prognostic factor for LR, as confirmed by a large review on more than 17,500 patients [12]. The risk of finding an infiltrated CRM increases with pathological stage [13]. The risk of 5-year LR is about 10% in mesorectal plane of TME, compared to 25% of recurrences in resections where the plane of TME is intramesorectal or along muscularis propria [14]. Moreover, in pre-TME era there was a much higher percentage of LRs than distant recurrences [15]. In advanced stage rectal cancers, the use of neoadjuvant therapy plays an important role in avoiding recurrences. In literature, Various studies have reported significantly lower risk of LR in patients receiving neoadjuvant radiotherapy or chemoradiotherapy [16,17,18]. In a large study on patients with advanced stage rectal cancer, the LR rate at 5 years was 4.6% in those receiving neoadjuvant radiotherapy and 11% in those undergoing only surgery [13].

Another factor conditioning the risk of LR is the T parameter in the TNM classification: in T1-T2 patients recurrences occur in 1% of cases with negative CRM and 12% of cases with positive CRM; in T3-T4 patients the percentage rises to 15% with negative CRM and 25% with positive CRM. A tumour invading the anterior peritoneal reflection line (T4) is an independent risk factor for intraperitoneal recurrence [13].

High risk of LR is also related with tumor location in the lower 1/3rd of the rectum, presence of tumor budding, poorly differentiated tumor histology, positive nodal status, perineural spread, and extramural venous invasion [6]. The risk of LR is lower if the mesorectal excision ensures removal of all mesorectal lymph nodes [8]. In their study on 481 consecutive rectal cancer, Rasanen et al. confirmed that distal tumor location, positive CRM, mucinous histology and local transanal excision are factors associated with LR [9].

The pattern of LR in rectal cancer has an important role in the choice of operative approach and a prognostic significance. Therefore, several classification systems have been proposed for describing the pattern of LR in rectal cancer. The Memorial Sloan Kettering Cancer Center (MSKCC) has developed a classification dividing LRs in 4 types according to the location: axial, that refers to anastomotic or perineal involvement; anterior, involving the genitourinary structures; posterior, that refers to sacrum involvement and lateral, involving pelvic bone, muscles and soft tissues of the pelvic sidewall, and neurovascular structures [19]. Central/ axial recurrences according to MSKCC classification are associated with better prognosis compared to posterior or lateral recurrences [20] According to the Leeds classification the recurrences of rectal cancer are divided into central (pelvic structures without involving bone), sacral (involvement of the sacrum), sidewall and composite (both sacrum and sidewall are involved) [20]. The Beyond TME (BTME) Consensus Group recently proposed a MRI-based classification of rectal cancers that divides the pelvis into seven compartments formed by fascial boundaries along the potential planes of dissection: the Anterior above Peritoneal Reflection, the Anterior below Peritoneal Reflection, the central, the posterior, the lateral, the infra-elevator, and the anterior urogenital triangle compartment [21]. Georgiou et al. showed that tumours showing on MRI involvement of the “anterior above peritoneal reflection” (including the iliac vessels, sigma, small intestine, and lateral sidewall pelvic fascia) are linked to poorer survival than those in which there is no such involvement. Reduced disease-free survival occurs if three or more compartments are involved [22]. A recent review including a total of 58 studies and 3975 patients with locally recurrent rectal cancer classified according to BTME system reports that most recurrences occur in the central compartment (including the rectum and perirectal fat), followed by the lateral compartment (involving the external iliac vessels and lymph nodes, the piriformis muscle, and the internal obturator). Many cases of recurrence are reported in the posterior (involving the pre-sacral fascia, sacrum, and sciatic nerve) and anterior below peritoneal reflection (including the genitourinary structures and pubic symphysis) compartments, with much lower numbers or no LRs within the anterior above, anterior urogenital (including the urethra, the introitus vaginal, and the crus penis), and infra-elevator compartments (involving the elevator of the anus muscle and the external sphincter) [23].

Recurrences in central area result from intraoperative tumor spillage due to incomplete TME. This kind of recurrences where very common in the pre-TME era. Moreover, anastomotic leakage following anterior resection increases the risk of central LR [24]. Lateral and posterior LR are related with rectal tumors over 5 cm from the anal verge treated with neoadjuvant radio or chemiotherapy, because of the direct spread through the mesorectum to lateral pelvic sidewall lymph nodes, that are not resected during TME [25].

Treatment of LR is often complex. Several factors may be a contraindication for LR surgical resection, such as extensive lateral pelvic sidewall involvement, encasement of external iliac vessels, and the presence of distant metastases. However, today, improvements in surgical technique that allow a more aggressive approach mean that most of these contraindications are no longer absolute [26]. The study of Rasanen et al. demonstrated that curative treatment was possible in 25% of patients with local recurrence compared to 36.6% of patients with distant metastases [9]. Moreover, using a multimodality approach including neoadjuvant CRT, intraoperative radiation therapy and/or adjuvant chemotherapy can significantly improve prognosis, as demonstrated by Bird et al. in their study on 98 patients with locally recurrent rectal cancer [27].

## 3. Computed Tomography (CT)

Currently, Contrast-Enhanced (CE)-CT is the imaging modality of choice for staging rectal cancer and for follow-up, ensuring reproducibility of results for future comparison [4,6]. Surveillances strategies are aimed to identify LRs, distant metastasis and new metachronous neoplasms. Many meta-analysis and randomized trials comparing different surveillance strategies have reported that surveillance increase detection of total recurrences and potentially threatening recurrences [28,29,30]. ESMO guidelines recommended at least 2 CT scans of complete chest and abdomen in the first three years [6]. In accordance with the NCCN guidelines, patients with stage II-III should undergo CT examination every 6–12 month for at least 5 years while for patients staged IV a CT scan every 3–6 months for 2 years and then every 6–12 months for up to 5 years [4]. In contrast, the surveillance program proposed by the recent RESARCH study for patients enrolled in the watch and wait strategy includes chest and abdomen CT scans annually [31].

CT scan is recommended to detect potentially resectable metastatic lesions, primarily in the lung and the liver (Figure 1) [4,8]. Moreover, CT can detect extraluminal recurrence non-evident on endoscopy. However, due to its low soft-tissue contrast, CT is limited in evaluating local tumor recurrence within pelvis. In particular, the postoperative changes such as fibrosis can mimick LRs [32] A meta-analysis by Maas et al. explored the accuracy of whole-body imaging for patients with suspected local recurrence of RC; among PET, PET/CT, CT and MRI, CT showed the lowest diagnostic performance [33]. CT with multi-planar reformatting, now widely available, can partly overcome these limitations as was demonstrated by Stueckle et al., who found a sensitivity of 88%, a specifity of 98%, an accuracy of 96% and a positive predictive value of 94% for multiplanar reconstruction [34].

A non-contrast sequence is useful to better depict surgical changes and to obtain a panoramic view of the entire abdomen. A late arterial and portal phase images should be acquired respectively at 15–20 and 60–70 s after contrast injection; a delayed phase obtained after 5 min from the start of contrast media injection is mandatory to detect small implants of carcinomatosis and liver metastasis. Moreover, urographic post-contrast enhancement could be helpful to assess urinary involvement.

The first postoperative CT often shows treatment outcomes such as pre-sacral fluid collections or fibrotic changes; differentiation of recurrence from post-surgical changes on CE-CT after surgery may be challenging. Therefore, the radiologist must pay attention to the stability or reduction of these findings over time (Figure 2) [35,36].

A bowel wall thickening near the surgical clips, a soft-tissue attenuation nodule in the surrounding organs or enlarged regional lymph nodes represent findings of locoregional recurrence and CT helps in describing the LR pattern [37].

In addition, CT could be useful in some urgent clinical setting, i.e., in evaluating patients presenting with an acute abdomen with bowel obstruction that may be due to either surgical adhesions or to recurrent disease [37].

## 4. Magnetic Resonance Imaging (MRI)

NCC guidelines recommends MRI as primary surveillance imaging in patients treated with trans-anal local excision (every 3–6 months for 2 years then every 6 months for a total of 5 years), along with proctoscopy and endoscopic ultrasound [4]. For patient who have undergone TME, MRI is used as a problem-solving tool in patients with equivocal findings from other imaging techniques, such as CT or in symptomatic patient with a negative endoscopic exam [18]. Additionally, in patients with complete clinical response after chemoradiation therapy, MRI is also recommended by the recent RESARCH study as primary imaging surveillance during the “watch-and-wait” (at least every 6 months for the first 2 years) [31].

The recommended MRI protocol by the MERCURY study group (Magnetic Resonance Imaging and Rectal Cancer European Equivalence) includes two-dimensional (2D) fast spin echo (FSE) T2-weighted sequences without fat suppression. These sequences, using a small field of view and slice thinner than 3 mm, should be obtained in three planes: oblique axial plane, which is perpendicular to the tumor in order to avoid blurring of the muscularis propria; sagittal plane, which is longitudinal to the tumor axis; and oblique coronal plane, which is parallel to the anal canal and allows to evaluate the relationship with the anal sphincter [38]. FSE T2-weighted MRI without fat suppression but with a large field of view are recommended in the axial plane, allowing the evaluation of distant lymph node chains, and in the sagittal plane, allowing the localization of the recurrence, enabling the measurement of its height and its relationship to the midline structures. In addition, a high-resolution T2-weighted in oblique coronal plane, parallel to the anal canal, should be performed in rectal cancers that are near or involve the anal sphincter complex. T2-weighted sequences, in various studies, have showed sensitivity ranges between 72% and 88.9% and higher values of specificity, reaching 100% [39,40].

Concerning the use of spasmolytic agents and induction of endo-rectal distention there is no consensus, as well as for the addition of gadolinium-enhanced T1-weighted images and Diffusion Weighted Imaging (DWI) with a high b-value (≥800 s/mm^2^). However, according to some studies, the diagnostic performance of MRI for tumor restaging after CRT and for diagnosis of LR can be improved by gadolinium-enhanced MRI and DWI as compared to T2-weighted imaging alone [41,42,43,44,45,46,47].

In a retrospective study on 326 patients who underwent rectal cancer surgery, Lee et al. observed that MRI, both with or without contrast, has a statistically significant superior performance in detecting LR compared with CT in patients at a high risk of recurrence. Moreover, they concluded that the diagnostic accuracy of non-enhanced MRI is similar to that of contrast-enhanced MRI for the detection of pelvic recurrence [47]. This result is in contrast to the earlier one, carried out by Titu et al., who concluded that there is no difference between MRI and conventional follow-up tests in the ability to detect cases suitable for surgery, indeed, in their opinion, MRI use during surveillance is not justified [48]. However, these two studies differ in the sequences used; notably the study by Lee et al., included DWI, which has already been shown to increase the sensitivity in the detection of persistent tumor after neo-adjuvant therapy. Colosio et al., indeed, prospectively evaluated the potential of DWI MRI, demonstrating an increase in the sensitivity for the detection of persistent tumor compared with T2-weighted sequences alone (Figure 3) [44].

During surveillance, radiologist should pay attention to the primary tumor’s morphology and to T2 signal of the anastomotic region [40,49,50,51]. The main pitfall is the discrimination between tumor recurrence and post-surgery changes, such as thickened wall with high signal intensity on T2-weighted images reflecting granulation tissue, blood products, and radiation-induced changes [49]. Concurrently, fibrous changes can hide LR occurring in the context of a scar; in this context DWI may improve the accuracy of MRI in differentiating fibrosis, which show low intensity value on high *b*-value, and residualtumor demonstrating high signal intensity with lower apparent diffusion coefficient values (Figure 4) [52]. Some authors suggested the use of post-contrast sequences to discriminate the two entities, observing a greater enhancement and sometimes rim-shape enhancement for the LRs; however, accuracy of these observations have not been proved [49]. Molinelli et al., in their study highlighted how the addition of DWI or post-contrast T1 sequences improved diagnostic performance for LR diagnosis compared to T2-weighted sequence alone [41]. Similary, Lambregts et al., by comparing standard MRI + DWI with standard MRI protocol alone, found that the addition of DWI increased the sensitivity for LR diagnosis in patients who underwent preservation treatment (CRT+ trans-anal local excision or “watch-and-wait”) [53].

A mucin-like change visible as an area of high signal intensity on T2- weighted images can be misinterpreted as a seroma but could be a recurrent mucinous tumor [49].

Radiologist attention must then be directed to the surrounding fat plans. This is also a noteworthy non-specific finding, that may represent both tumor invasion of the adjacent structures or post-treatment changes. Another potential pitfall, observed by Sinaei et al., is the dislocation of pelvic structures in the rectal fossa, particularly seminal vesicles, prostate and uterus or ovaries [49].

When a suspected recurrence occurs, it is important, for the management, to report the possible Presacral involvement above the S1 or S2, involvement of the sciatic nerve, of the external or common iliac vessel, and of the pelvic side bones; all these findings may contraindicate surgical treatment [54].

Radiomics analysis may be a valid tool to overcome diagnostic imaging limitations [55,56,57,58,59,60]. Radiomics analysis is an emerging technique based on the extraction of quantitative data from medical images. These objective data, referred to as radiomic features and not visible to the human eye, are analysed in order to construct a radiomic model capable of more accurate tumor diagnosis, staging and prognosis. Recently, the research in rectal cancer focused mainly on radiomic-based on MRI since it showed more potential and significance than radiomic-based on CT or PET/CT. Recently, Chen et al., tried to test radiomics in differentiating LR from anastomotic non-recurrence lesions. In their study 80 patients, with a suspected clinical recurrence, underwent a 3.0T MRI. Radiomic features were extracted from the volume of interest and then used in order to build four radiomic models based on T2W sequences, DWI sequences, contrast-enhanced T1W sequences and on the combination of the three mentioned sequences. Good results have been achieved in terms of sensitivity and specificity by the combination model whit values of 81.82% and 75.86% respectively in the validation cohort [61]. In addition, radiomics may be a useful tool to predict clinical outcomes or prognostic features. Jayaprakasam et al. in their study on 236 patients with locally advanced rectal cancer who underwent neoadjuvant chemoradiotherapy demonstrated that radiomics analysis of mesorectal fat can predict local recurrence with a sensitivity and specificity of 68.3% and 80.7% rescpectively [62]. Petkovska et al., demonstrated that T2-radiomics score combined with anatomical MRI model may accurately differentiate between the responders and non-responders to neoadjuvant CRT [63]. In this view, radiomics could help to stratify patients with locally advanced rectal cancer before treatment and direct toward more individualized therapies. However, more studies with larger populations are needed to further validate these results.

## 5. Functional Imaging

The routine use of PET-CT during post-operative surveillance is not recommended by both ESMO and NCCN guidelines. However, the NCCN guidelines suggest its employment as problem solving tool in patients undergoing surveillance or after a LR diagnosis with other detection modality, such as increased CEA level or positive CT scan, with the aim of detect other disease sites [4,8]. 18F-fluorodeoxyglucose (FDG)-PET proved, in a large systematic review carried out by Lu et al., a sensitivity and specificity for LR of 94.1% (95% CI, 89.4–97.1%) and 77.2% (95% CI, 66.4–85.9), respectively [64]. A systematic review and meta- analysis found 165 article that addressed the performance of CT, MRI and PET-TC on the detection of colorectal liver metastases; a significant higher sensitivity was found for the functional modality [65].

Despite these evidence, PET/CT role remain marginal due to its limitations in detecting small recurrences; furthermore, inflammatory changes may result in false positives and false negatives may occur due to mucinous tumors recurrence [6].

In order to overcome these limitations, among the functional imaging techniques, MRI-PET is showing promising results, thanks to its better soft tissue contrast. Using the combination of MRI accuracy and the RC’s FDG avidity, MRI-PET proved a sensitivity in detecting LR of 94%, specificity 88%, a positive/negative predictive value of 97% and 78% respectively and accuracy 93%. In the recent study carried out by Plodeck et al., FDG-PET/MRI showed higher values of sensitivity and specificity than MRI [66]; this novelty suggests a possible role of MRI-PET in the setting of RC follow-up. Moreover, FDG-PET/MRI showed an accuracy of 100% in detecting residual disease after neoadjuvant CRT compared to 71% of MRI, so that FDG-PET may play a role during surveillance programs of patients enrolled in watch and wait strategy [67].

## 6. Conclusions

Although the incidence of locally recurrent rectal cancer has significantly decreased thanks to the advances in surgical and chemoradiation therapy, local recurrences are still a cause of morbidity and mortality. An early detection of LR improves prognosis and patient’s quality of life so that imaging plays a key role during follow up of rectal cancer. CT and MRI are the main surveillance imaging techniques recommended in these patients. However, CT lacks the soft-tissue resolution so that it shows a low diagnostic performance for locally recurrent rectal cancer and its use is limited to the identification of distant metastases. PET/CT is more accurate than CT and it is generally used for problem solving. MRI, instead, has an essential role in the surveillance and diagnosis of local recurrence due to its high soft-tissue resolution. Moreover, advanced MRI techniques, such as DWI, and Radiomics analysis may improve MRI diagnostic performance for LR diagnosis.

## Figures and Tables

**Figure 1 diagnostics-13-02104-f001:**
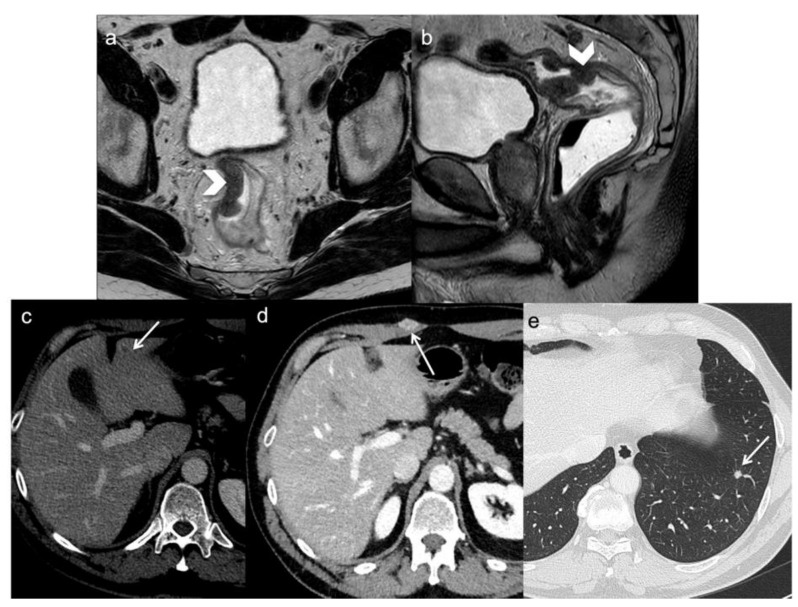
A 54-year-old man with rectal cancer undergoing anterior resection. FSE T2-weighted MRI obtained in oblique axial (**a**) and sagittal plane (**b**) demonstrate the primary tumor (arrows head). During the follow-up, CT scan demonstrates distant metastasis (arrows) two years following surgery: a small hepatic recurrence (**c**), a skin (**d**) and a lung metastasis (**e**).

**Figure 2 diagnostics-13-02104-f002:**
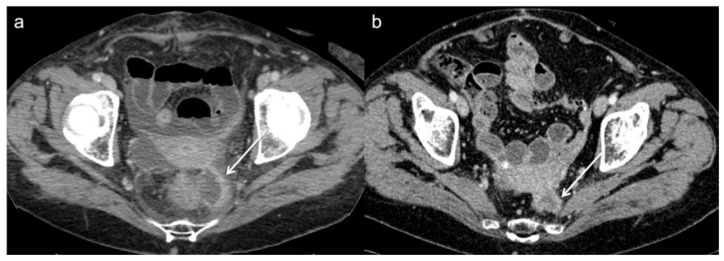
Post operative CT scan of a 76 year-old woman that underwent Hartmann’s procedure. (**a**) Axial image from initial post-operative baseline surveillance CT demonstrates fluid collection and fibrotic changes near to the anastomotic region (arrow). (**b**) These findings was reduced on follow-up CT in keeping with post-operative changes.

**Figure 3 diagnostics-13-02104-f003:**
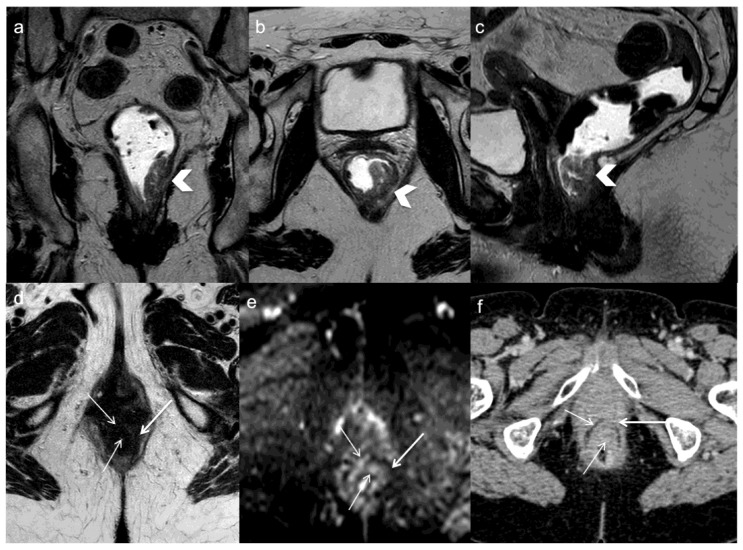
A 71 years old woman with lower rectal cancer undergoing neoadjuvant chemoradiotherapy and anterior resection. FSE T2-weighted MRI obtained in coronal (**a**), oblique axial (**b**), and sagittal plane (**c**) demonstrate the primary tumor (arrows head). One year after surgery oblique axial T2-weighted image (**d**) demonstrates concentric parietal lesion with low and intermediate signal intensity and restricted diffusion on DWI (arrows) (**e**). CE-CT demonstrates recurrent tumor which is not not clivable anteriorly from the vagina (arrows) (**f**).

**Figure 4 diagnostics-13-02104-f004:**
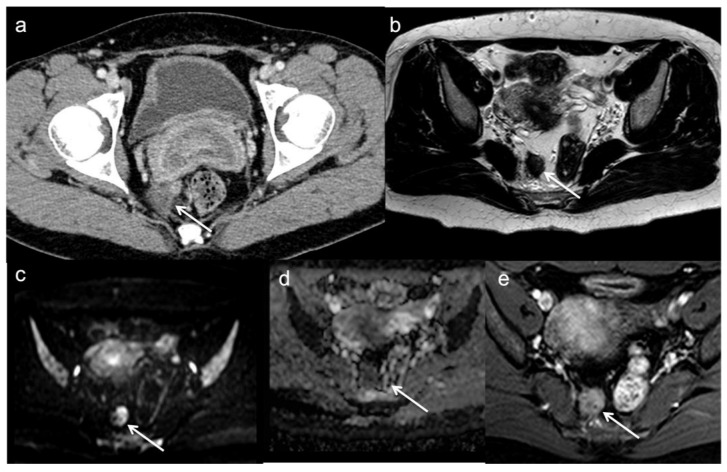
Local recurrent tumor one year after anterior resection. CE-CT image (**a**) demonstrates an irregular mass (arrow) in the fat near to the anastomotic region. MRI confirms the local recurrence (arrows) with low T2 signal (**b**), restricted diffusion on DWI (**c**), low apparent diffusion coefficient values (**d**) and enhancement on gadolinium-enhanced T1-weighted image (**e**).

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
