# Peer review of "Local Recurrences in Rectal Cancer: MRI vs. CT"

_diagnostics, 2023, doi:10.3390/diagnostics13122104_

Round 1
Reviewer 1 Report
I would like to thank the authors for the review and synthesis effort on the role of imaging in the early detection of local recurrence due to rectal cancer.
I would like to make some suggestions to improve the composition of the article and the usefulness of the work.
Maior issues
The purpose of the authors is twofold: 1) to analyze the risk of local recurrence and the pattern of local recurrence of rectal cancer and 2) to review the role of imaging in the early detection of local recurrence. The first objective is addressed in paragraph 2, and the second objective in paragraphs 3-5. While the objectives are clear, the methodology employed is not. In fact, there is no methodology section to explain the scope of the review (what clinical situations, what treatments), nor how the literature review has been carried out or how the results have been analyzed.
On the other hand, the text adopts a teaching style in some paragraphs (entire section 2, lines 169-178, lines 237-245), provides technical recommendations in others (lines 194-208, 245-253), or provides advice to avoid making mistakes in the interpretation of the images (lines 262-266). The article includes considerations for patients who have received both surgical and non-surgical (lines 186-193), while considering aspects of radiomics and functional imaging with PET. The result is variegated and heterogeneous, hindering an orderly presentation by dividing the article into sections by therapeutic modalities instead of by clinical situations.
I would suggest to define the methodology of the review and to consider the organization of the article according to clinical situations so that the presentation is clearer and more orderly.
Minor issues
Line 33: update NCCN guidelines to the most recent one (2023)
Line 40: reference 5 is not a guideline as it was supposed to be because the text says “most guidelines recommend…).
Lines 43-48: NCCN and ESMO guidelines recommend colonoscopy in the follow-up, but it is not mentioned.
Line 50: the convenience of LR early detection, being intuitive, is not supported by the provided data. Nevertheless, it is concluded (line 321-322) that “An early detection of LR improves prognosis and patient’s quality of life”. Conclusions should be supported by data. Therefore, data should be added of conclusion should be removed.
Line 65: Reference 9 includes only colon cancer patients. It should be mentioned or removed.
Line 133-137: For the treatment of local recurrence there are different therapeutic options, but only surgical salvage is mentioned.
Line 142: The “many meta-analysis and randomized trials” sentence is supported by reference 27. Check, because that paper is a dosimetric study.
Line 153: The Maas meta-analysis was published in 2011. As the authors mentioned before that “many meta-analysis” have been reported, the most recent meta-analysis should be provided.
Line 155: The Stueckle study is from 2005. Look for a more recent study.
Lines 163-168: The authors recommend to obtain four separate series of CT images: non-contrast, arterial phase, portal phase, and a mandatory delayed phase. I could not find that recommendation in the supporting reference 18.
Lines 186-193: The clinical situations described in this paragraph (surveillance after local excision and surveillance during the “watch and wait” strategy, are not described in the introduction and not mentioned in other sections. It should be clear from the introduction if those clinical situations are included in the review. Reference 16 does not support any recommendation regarding imaging after chemo-radiation therapy complete clinical response.
Figure 3: sagittal (c)-coronal (a); Images d-e-f are not clearly demonstrative of recurrent tumor. The findings are very subtle.
Author Response
Reviewer 1
I would like to thank the authors for the review and synthesis effort on the role of imaging in the early detection of local recurrence due to rectal cancer.
I would like to make some suggestions to improve the composition of the article and the usefulness of the work.
Maior issues
The purpose of the authors is twofold: 1) to analyze the risk of local recurrence and the pattern of local recurrence of rectal cancer and 2) to review the role of imaging in the early detection of local recurrence. The first objective is addressed in paragraph 2, and the second objective in paragraphs 3-5. While the objectives are clear, the methodology employed is not. In fact, there is no methodology section to explain the scope of the review (what clinical situations, what treatments), nor how the literature review has been carried out or how the results have been analyzed.
On the other hand, the text adopts a teaching style in some paragraphs (entire section 2, lines 169-178, lines 237-245), provides technical recommendations in others (lines 194-208, 245-253), or provides advice to avoid making mistakes in the interpretation of the images (lines 262-266). The article includes considerations for patients who have received both surgical and non-surgical (lines 186-193), while considering aspects of radiomics and functional imaging with PET. The result is variegated and heterogeneous, hindering an orderly presentation by dividing the article into sections by therapeutic modalities instead of by clinical situations.
I would suggest to define the methodology of the review and to consider the organization of the article according to clinical situations so that the presentation is clearer and more orderly.
We made the recommended changes. In particular, we reduced the teaching style of section 2. In the introduction, we specified that the main purpose of this review (as the title suggests) is to compare rectal cancer surveillance imaging methods to analyze which one is best in identifying local recurrences. Therefore, we decided to leave the organization of the text based on imaging methods rather than clinical situations.
Minor issues
Line 33: update NCCN guidelines to the most recent one (2023)
The most recent NCCN guidelines for rectal cancer were published in 2022 and reference is the 4.
Line 40: reference 5 is not a guideline as it was supposed to be because the text says “most guidelines recommend…).
Yes, we confirmed.
Lines 43-48: NCCN and ESMO guidelines recommend colonoscopy in the follow-up, but it is not mentioned.
We added colonoscopy
Line 50: the convenience of LR early detection, being intuitive, is not supported by the provided data. Nevertheless, it is concluded (line 321-322) that “An early detection of LR improves prognosis and patient’s quality of life”. Conclusions should be supported by data. Therefore, data should be added of conclusion should be removed.
LRs represent a significant clinical problem causing debilitating symptoms such as bleeding, pelvic pain, and fistula. In addition, treatment of LR is often complex. For these reasons, “An early detection of LR improves prognosis and patient’s quality of life”.Therefore, also Ganeshan et al. in their work concluded “Imaging
surveillance for detection of locally recurrent rectal cancer would potentially lead to early surgical intervention, optimal surgical planning, decrease in post-surgical complications, and ultimately may improve patient’s quality of life.“ [Ganeshan, D.; Nougaret, S.; Korngold, E.; Rauch, G.M. and Moreno, C.C. Locally recurrent rectal cancer: what the radiologist should know. Abdom Radiol, 2019, 44, 3709–25. https://doi.org/10.1007/S00261-019-02003-5]
Line 65: Reference 9 includes only colon cancer patients. It should be mentioned or removed.
Ok, we removed reference 9
Line 133-137: For the treatment of local recurrence there are different therapeutic options, but only surgical salvage is mentioned.
We added other different therapeutic options for local recurrence
Line 142: The “many meta-analysis and randomized trials” sentence is supported by reference 27. Check, because that paper is a dosimetric study.
We corrected this error
Line 153: The Maas meta-analysis was published in 2011. As the authors mentioned before that “many meta-analysis” have been reported, the most recent meta-analysis should be provided.
We have not found more recent studies that specifically address this topic
Line 155: The Stueckle study is from 2005. Look for a more recent study.
We have not found more recent studies that specifically address this topic
Lines 163-168: The authors recommend to obtain four separate series of CT images: non-contrast, arterial phase, portal phase, and a mandatory delayed phase. I could not find that recommendation in the supporting reference 18.
We removed the reference
Lines 186-193: The clinical situations described in this paragraph (surveillance after local excision and surveillance during the “watch and wait” strategy, are not described in the introduction and not mentioned in other sections. It should be clear from the introduction if those clinical situations are included in the review. Reference 16 does not support any recommendation regarding imaging after chemo-radiation therapy complete clinical response.
We corrected the reference error and added a description of watch and wait strategy in the introduction, and in section 3 and 5
Figure 3: sagittal (c)-coronal (a); Images d-e-f are not clearly demonstrative of recurrent tumor. The findings are very subtle.
Thank you for pointing out the error sagittal/coronal. We tried to emphasize the small recurrence by adding more arrows
Reviewer 2 Report
Several case studies have been provided to show the effectiveness of CT/MRI in revealing rectal cancer pathologies.
1. The authors should revise the abstract to follow the IMRAD structure.
2. The conclusion should elaborate more on the findings, instead of medical imaging comparison.
-
Author Response
Reviewer 2
Several case studies have been provided to show the effectiveness of CT/MRI in revealing rectal cancer pathologies.
- The authors should revise the abstract to follow the IMRAD structure.
We revised the abstract as suggested.
- The conclusion should elaborate more on the findings, instead of medical imaging comparison.
We apologize because perhaps the purpose of our review was not clear. For this reason, in the introduction, we specified that the main purpose of this review is to compare rectal cancer surveillance imaging methods analyzing which one is best in identifying local recurrences.